# Cardiovascular Risk Profiles and Pre-Existing Health Conditions of Trekkers in the Solu-Khumbu Region, Nepal

**DOI:** 10.3390/ijerph192416388

**Published:** 2022-12-07

**Authors:** Miriam Haunolder, Christian Apel, Daniela Bertsch, Carina Cerfontaine, Michael van der Giet, Simone van der Giet, Maren Grass, Nicole Maria Heussen, Nina Hundt, Julia Jäger, Christian Kühn, Sonja Musiol, Lisa Timmermann, Knut Wernitz, Ulf Gieseler, Audry Morrison, Volker Schöffl, Thomas Küpper

**Affiliations:** 1Institute of Occupational, Social & Environmental Medicine, RWTH Aachen Technical University, 52074 Aachen, Germany; 2Medical Department, St. Antonius Hospital, 52249 Eschweiler, Germany; 3Department of Dental Preservation, Parodontology & Preventive Dentistry, RWTH Aachen Technical University, 52074 Aachen, Germany; 4Institute of Applied Medical Engineering, RWTH Aachen University, 52074 Aachen, Germany; 5Department of Internal Medicine & Cardiology, Ilmtalklinik, 85276 Pfaffenhofen, Germany; 6Department of Medical Statistics, RWTH Aachen Technical University, 52074 Aachen, Germany; 7Medical School, Sigmund Freud Private University, 1020 Vienna, Austria; 8Medical Commission of the Union Internationale des Associations d’Alpinisme (UIAA MedCom), 3007 Bern, Switzerland; 9Royal Free London NHS Foundation Trust, London NW3 2QG, UK; 10Department of Sport Orthopaedics, Klinikum Bamberg, 96049 Bamberg, Germany; 11Department of Trauma Surgery, University Hospital Erlangen-Nuremberg, 91054 Erlangen, Germany; 12School of Applied and Clinical Sciences, Leeds Becket University, Leeds LS2 9JT, UK; 13Section of Wilderness Medicine, Department of Emergency Medicine, University of Colorado School of Medicine, Denver, CO 80045, USA

**Keywords:** cardiovascular risk factors, high altitude, coronary heart disease, arrhythmia, trekking, acute mountain sickness

## Abstract

Background: High-altitude tourist trekking continues to grow in popularity on the Everest Trek in Nepal. We examined which pre-existing cardiovascular and health conditions these global trekkers had and what health issues they encountered during the trek, be it exacerbations of pre-existing conditions, or new acute ones. Method: Trekkers (*n* = 350) were recruited from guesthouses along the Everest Trek, mostly at Tengboche (3860 m). After completing a questionnaire on their health and travel preparation, they underwent a basic physical examination with an interview. Results: Almost half (45%) had pre-existing conditions, mostly orthopaedic and cardiovascular diseases. The average age was 42.7 years (range 18–76). The average BMI was 23.4 kg/m^2^, but 21% were overweight. A third were smokers (30%), and 86% had at least one major cardiovascular risk factor. A quarter (25%) were suffering from manifest acute mountain sickness (AMS), and 72% had at least one symptom of AMS. Adequate pre-travel examination, consultation, and sufficient personal preparation were rarely found. In some cases, a distinct cardiovascular risk profile was assessed. Hypertensive patients showed moderately elevated blood pressure, and cholesterol levels were favourable in most cases. No cardiovascular emergencies were found, which was fortunate as timely, sufficient care was not available during the trek. Conclusion: The results of earlier studies in the Annapurna region should be revalidated. Every trekker to the Himalayas should consult a physician prior to departure, ideally a travel medicine specialist. Preventative measures and education on AMS warrant special attention. Travellers with heart disease or with a pronounced cardiovascular risk profile should be presented to an internal medicine professional. Travel plans must be adjusted individually, especially with respect to adequate acclimatisation time and no physical overloading. With these and other precautions, trekking at high altitudes is generally safe and possible, even with significant pre-existing health conditions. Trekking can lead to invaluable personal experiences. Since organized groups are limited in their flexibility to change their itinerary, individual trekking or guided tours in small groups should be preferred.

## 1. Introduction

Cardiovascular disease (CVD), hypertension, diabetes, and death due to cardiovascular events are on the rise in modern society [1,2]. At the same time, tourism is the world’s biggest industry [3]. The growing popularity of high-altitude trekking attracts a broad spectrum of novice and experienced trekkers and alpinists to destinations such as the Everest Basecamp Trek in Nepal. Comprehensive, all-inclusive trekking package tours to remote mountainous areas like this are now readily available. More recently, these adventure holidays have attracted a new type of traveller: those seeking the adventure of a lifetime in the mountains without any experience of high altitudes or trekking [3]. Our objective was to examine what cardiovascular risk factors or pre-existing health conditions the trekkers on the Everest Basecamp Trek had and what health problems they encountered—i.e., exacerbations of existing conditions and/or new acute symptoms.

Although results differ from region to region, such data are available for the European Alps [4,5], or the Pyrenees [6], where they indicate a quite low rate of incidences., e.g. Faulhaber No data were found on the presence of cardiovascular risk factors. However, trekking typically takes place at significantly higher altitudes, which potentially triggers the risk of hypoxic pathomechanisms. Trekkers typically go to a very remote environment, which itself increases the risk of fatal outcome if a significant incident should occur. Data on cardiovascular risk factors and disease in trekkers in the Himalayas are scarce. Only fatalities in trekkers en route to the summit of Mt. Everest are somewhat documented [7,8] and case reports are relatively rare [9,10]. Cardiac symptoms are easily confused with signs of acute mountain sickness (AMS) and vice versa. Gathering more information about the people potentially at risk because of underlying diseases or risk factors in such a setting is valuable for example to the practitioners at home, trekking organisations and local helpers, and of course to the individuals trekking in the Himalayas themselves. The present study focuses on trekkers in the Solu-Khumbu region, Nepal, a region with growing numbers of visitors and with the opportunity for a great trekking adventure in one of the most scenic landscapes of the world.

## 2. Materials and Methods

Study design: The epidemiological study at hand was but one project within the ADEMED (Aachen dental and medical expedition) to the Everest Region in Nepal, profiting of the experiences of the former ADEMED 2008 to the Annapurna Region and further evaluating the information gathered there.

Trekkers on Everest Basecamp Trek, Nepal, were asked to complete a questionnaire with over 100 items which consisted of yes or no answers and optional free-text entries. It was available in German, English, and French languages. Data were collected by the author of this studies and two of the co-authors. They assisted participants in the cases of language barriers.

Recruitment of participants: The science posts were situated outside guest houses along the main trekking route to Everest Base Camp, first in Gorak Shep (altitude 5160 m), then in Dingboche (4360 m), Tengboche (3680 m), Namche Bazar (3440 m), and Lukla (2800 m). Participants were either contacted directly when passing by or attracted by the posted signs. The response rate was very good and estimated at about 80%. Whole groups of trekkers would participate as well as individuals, and breakfast and dinner times in the guest house were also valuable opportunities to have a whole room of trekkers participate in the study.

Questionnaire and physical examination: The questionnaire concerned basic health information, previous medical history, medication, drug use, lifestyle and travel preparations (e.g., vaccinations, consultations, fitness training), as well as health issues encountered en route, especially cardiac symptoms and those of acute mountain sickness (AMS). Participants underwent a basic physical examination, including measurements of blood pressure (manually via Korotkow method), heart rate, and oxygen saturation via a fingertip pulse oximeter (Pulsoxymeter CMS 50E). Body weight and height were anamnestic information and the basis for body mass index (BMI) calculation. A digital set of scales (Soehnle Sense Comfort 100, Leifheit AG, Nassau/Germany) measured each participant’s current weight (clothed but without boots and jacket) and weight of the backpack. Total cholesterol was measured as fasting levels in the morning (Accutrend^®^ Cholesterol by Roche) and thus only available for a limited number of trekkers.

Data analysis: Data analysis was performed in Excel, significance was calculated via Chi-Square Test, Fisher’s Exact Test, and U-Test (*p* ≤ 0.05). Presence of AMS was assessed using a symptom-orientated clinical score and the Lake Louise Score of 1993 [11]. An analysis with the adapted Lake Louise Score of 2018 was retrospectively conducted [12]. A healthy lifestyle was defined as >4 healthy items (e.g., BMI < 25, non-smoker, 3 or more hours of physical activity per week) according to the “healthy lifestyle score” of the University of Tasmania [13]. Heart risk and heart age were calculated according to the data sheets of the Framingham Heart Study [14]. This required either the participant’s BMI or cholesterol parameters in addition to their sex, age, systolic blood pressure (treated or untreated), smoking status (smoker/non-smoker), and diabetic disease (yes/no). The BMI-based version was used for all participants; if a cholesterol test was available, the cholesterol-based formula was used additionally. Major and minor risk factors were defined according to a commonly used definition of coronary heart disease and its risk factors [14].

Exclusion criteria: Due to the advice of the ethical commission, persons younger than 18 years of age were excluded. We did not discuss this topic with the commission further since there would be no relevance; the effect of cardiac risk factors at that age is neglectable. The study was designed according to the Declaration of Helsinki and consulted by the ethical commission of RWTH Aachen Technical University (Study no. Ek 196/11).

## 3. Results

Basic data and origin: We recruited 350 trekking volunteers from guest houses along the Everest Trek, mostly (*n* = 201, 57%) at Tengboche (3860 m). The volunteers were predominantly of Caucasian ethnicity (*n* = 302, 86%), and 61% were male (*n* = 211). The trekkers were largely from Europe (*n* = 237, 68%), with French (*n* = 50, 14%) and British (*n* = 40, 11%) being the most frequent nationalities. The average age was 42.7 years (range 18–76 years). The majority of trekkers had a normal body mass index (BMI) averaging 23.4 kg/m^2^ with 8 individuals classified as obese (2%), and 75 were overweight (21%).

Itinerary and travel preparations: Trekkers seldom travelled alone on the Everest Trek (*n* = 26, 7%). Half travelled in small groups of two to four people (*n* = 176, 50%). The average group size was 5.2 persons (range: 1–25). The mean duration of the ascent to Gorak Shep at 5160 m was 8.3 (2–37) days, and 51% of participants (*n* = 178) confirmed that they had taken at least one rest day for altitude acclimatisation. Prior to travelling, 92% of participants (*n* = 323) had informed themselves about their destination, although 69% (*n* = 243) admitted that they had not especially prepared for their trekking tour physically, and only 12% (*n* = 42) had prepared themselves by mountain or alpine tours. A quarter of the participants (*n* = 90, 26%) acknowledged that they were entirely inexperienced in trekking, and also a quarter of all trekkers (26.3%, *n*= 91) had not done any fitness training at all. Retrospectively, 5% (*n* = 17) wished they had been physically better prepared for their trekking experience.

Pre-travel medical advice: Half of the participants (*n* = 178, 51%) had seen a physician, and in most of these cases (49%), this was a visit to their general practitioner. Vaccination status before departure was 78% (*n* = 273). With respect to medication, 71% (*n* = 246) carried emergency medication, and 13% (*n* = 47) carried prescribed medication in their backpacks due to a pre-existing medical condition.

Acute symptoms: A third (33%) of the volunteers confirmed that unexpected events or incidents occurred during their trekking tour, mainly of a medical nature. The only non-medical incidents that clouded the trekking experience for some included snowfall, theft, and leech infestation.

When asked what discomforts they suffered during the trek, pain was identified by 120 respondents (34%), with a headache being the most frequently mentioned source (*n* = 63, 18%); the associated headache pain reached an average of 3.5 points on a scale from 1 to 10. Women were significantly more affected by pain than men (*p* < 0.05). Diarrhoea was mentioned by 62 participants (18%). Only 16 trekkers suffered from an injury (5%), mostly involving minor wounds to the lower limbs such as blisters.

Acute mountain sickness (AMS): At least one potential symptom of AMS was found in 61% (*n* = 215) of the subjects, and 25% suffered from manifest AMS with a symptom-orientated score of ≥3 points; the women were significantly more often affected compared to men (*p* < 0.001). Analysis using the Lake Louise Score of 1993 (where headache was counted as an obligatory symptom) revealed an AMS rate of 12.3% (*n* = 43). When retrospectively eliminating ‘sleep disturbance’ as a symptom for the Lake Louise Score of 2018, the collective showed 20 cases (5.7%) of AMS, and once again, the female sex was affected significantly more often (*p* < 0.01). Risk factors for high-altitude pulmonary oedema (HAPE) (cyanosis, fever, angina pectoris, heart rate > 100 bpm, oxygen saturation <90%, and dyspnoea at rest or during light activity) in combination with headache and at least one other AMS symptom were found in 30 trekkers (9%), and 19 (5%) trekkers showed signs of high-altitude cerebral oedema (HACE) such as headache in combination with nausea, dizziness, ataxia, or altered mental status. Medical aid was offered to everyone who seemed clinically unfit.

Cardiovascular symptoms and parameters: Shortness of breath was mentioned by 136 respondents (39%), which occurred most frequently under hard physical strain (*n* = 98, 28%). There was no difference in mean oxygen saturation between participants with or without dyspnoea, but in trekkers with dyspnoea, it occurred less often (−2.6%) and showed values above 90%. Peripheral oedema was present in 14 participants (4%). Heart palpitations were felt by 62 trekkers (18%), but only 5 of these were actually tachycardic. In total, there were 49 participants (14%) with a heart rate (HR) above 100 bpm at rest, and there was a tendency towards rising HR with increasing altitude (HR_mean_ 70.1 bpm in Lukla at 2800 m (42–96) vs. HR_mean_ 84 bpm in Tengboche at 3860 m (46–120), *p* < 0.01). Eight trekkers (2%) had experienced chest pain. Symptoms of cardiac insufficiency such as dizziness, weakness, and dyspnoea were mentioned by 79 participants (23%). In a quarter of all trekkers (*n* = 89, 25%), the fingertip reperfusion time was pathological (>2 s).

Mean oxygen saturation was 90.7% and mean SaO_2_ declined with increasing altitude (SaO_2 mean_ 95.1% (90–98) in Lukla at 2800 m versus SaO_2 mean_ 81.6% (78–88) *p* < 0.05 in Gorak Shep at 5140 m). Participants with current hypertensive blood pressure (BP) or previously known hypertension had a tendency towards slightly lower SaO_2_ values (n.s.).

Mean systolic BP was 128 mmHg (90–200), diastolic 79 mmHg (50–110) (Figure 1). Feasting cholesterol level was measured in 104 trekkers and showed an average of 178 mg/dL. The highest measured value was 288 mg/dL, the lowest was 150 mmHg, and in 68 tests, the result was ‘low’ (<150 mg/dL). Based on body mass index (*n* = 274), and additionally the cholesterol level in some cases (*n* = 32), the 10-year heart risk—meaning the risk of suffering a major cardiovascular event within the next 10 years—was calculated as an average of 6.9% (min. 1.0, max. 30.0%) with a mean heart age of 47.7 years (30–80 years), which was 5 years older than the actual mean age of the collective (Table 1). The difference between the BMI-based and the cholesterol-based formulas was 1.8% for heart risk in men (min. −1.4%, max. +3.9%) and 0.6% in women (min. −1.9%, max. 4.9%). Concerning heart age, the differences between the calculations were 2 years in men (min. −5 years, max. +5 years) and 1.1 years in women (min. −10 years, max. +9 years). The maximum difference between current age and heart age was +27 years in a 51-year-old participant from India suffering from obesity, hypertension, and nicotine addiction.

Pre-existing conditions: Almost half (*n* = 156, 45%) of the trekkers had pre-existing conditions, mostly orthopaedic (*n* = 91, 26%) and cardiovascular diseases (*n* = 42, 12%) (Figure 2). When asked for what discomfort they experienced during their trekking, eight (2%) stated that their known medical condition had worsened while trekking, but only two stated that this significantly affected their journey. Refer to the parallel study [15] for further details on conditions other than cardiovascular disease. Another 29 participants (8%) described that their pre-existing medical condition did not affect them during the trek.

Cardiovascular disease: Among the pre-existing cardiovascular conditions were 28 cases of hypertension (8%). The mean blood pressure among these participants was 149 mmHg systolic (100–180) and 86 mmHg diastolic (60–110), and 17 of them had acknowledged that they were on antihypertensive medication (mostly AT-1-receptor antagonists such as Lorsartan, *n* = 6). Other pre-existing cardiovascular conditions mentioned were hypotension (*n* = 1), right bundle branch block (*n* = 2), tachycardia (*n* = 1), bradycardia (*n* = 1), and one case of arrhythmia not further specified. None of the respondents claimed to have a coronary artery disease, but some took medication that hinted at a cardiovascular disease (cholesterol inhibitors (*n* = 3, 0.9%); acetylsalicylate acid (*n* = 2, 0.6%)). Three participants (0.9%) had suffered from a thrombosis in their past.

Almost a third of trekkers (*n* = 104, 30%) stated that their heart had been examined in the past (though not necessarily related to their trekking experience), mostly by electrocardiography (*n* = 53) or echocardiography (*n* = 19). An inconspicuous result was mentioned in 34 cases, and among those with abnormalities (*n* = 11), the following findings were stated: extra systoles, deviant heart axis, right bundle branch block, cardiomegaly, left-ventricular pathology, dysrhythmia, and accumulation of fluid in the lungs.

The vast majority of the trekkers (*n* = 301, 86%) had at least one major cardiovascular risk factor (Figure 3), with males more frequently affected (*n* = 211, 60%). Thirty percent (*n* = 104) were smokers, with an average of 12.5 pack-years. Only 17 trekkers were free of minor cardiovascular risk factors (5%) (Figure 4). Unfortunately, it was not possible to measure lipid subunits (HDL, LDL) in the field. Stress was defined as occupational stress, but emotional stress was not evaluated.

## 4. Discussion

The interviewed trekkers were older and more afflicted by pre-existing diseases than one would expect, especially when facing the physical challenge of an ascent to Everest Base Camp. They were significantly older than a collective previously investigated in the Alps [4,16] but of similar age compared to previous studies in the Annapurna region [17,18,19]. From our collective, 45% mentioned at least one pre-existing disease, and from this group, 70% had seen a general practitioner prior to departure to Nepal, but rarely a specialist was consulted (*n* = 1). In most cases, the check-up was likely with a medical provider lacking sufficient expertise in travel and high-altitude medicine.

Personal preparation was inadequate in a large number of participants, too. A quarter of the trekkers had not done any physical training prior to departure, and also, a quarter had never been in the mountains before, the Everest Trek being their first high-alpine experience ever.

In particular, individuals with pre-existing conditions were insufficiently prepared. For example, none of the three diabetics carried equipment for blood glucose testing and none of the participants with known hypertension (*n* = 28, 8%)—some of them under triple antihypertensive therapy—had any means of measuring blood pressure or any emergency drugs with them.

Only 24% of the participants had been vaccinated against rabies, which is recommended for Southeast Asia (especially Nepal and India) [20,21].

Thus, adequate pre-travel examination, consulting, and sufficient personal preparation were rarely found, which is alarming considering the growing and aging population of trekkers in destinations such as the Himalayas or other high-alpine settings and could be expected to lead to a large number of medical incidents or other impairments of the trekking experience. However, despite inadequate specialist pre-travel advice being given, the rate of acute medical incidents was low, and most trekkers were content with their journey.

Signs of AMS were present in 25% of the participants and manifest AMS according to Lake Louise Score in 12.3%, which was more than in previous studies [17,18,19]. However, AMS rates are generally declining. This decline can be explained by a growing tendency towards self-medication—e.g., 17% of our collective used acetazolamide. This self-medication can mask lesser AMS symptoms and may raise the number of the dangerous forms of AMS such as HACE and HAPE. The latter can hardly be distinguished from an acute or decompensated cardiac condition by lay persons except when they know and test the characteristic symptom “ataxia”.

In some cases, a distinct cardiovascular risk profile was assessed, and cardiovascular health conditions were the second most frequent pre-existing conditions among the collective. Hypertensive patients showed moderately elevated blood pressure, and cholesterol levels were favourable in most cases due to increased physical activity. No cardiac events such as acute coronary syndrome were witnessed during the study, but the combination of cardiovascular risk factors, unaccustomed physical strain, and decreasing oxygen availability certainly represents an explosive mixture of factors that could lead to any form of a cardiovascular emergency. Sufficient medical care would not be available in time given the remote setting of the Everest Trek, and studies have shown that emergency care provided by other trekkers, guides, or porters would not be realisable due to lack of knowledge [19]. Differentiation of severe AMS to other diagnoses is hardly possible.

This combination of various external and internal risk factors necessitates a qualified pre-travel check and individual advice. This should include cardiological aspects and those of high-altitude medicine, especially for any patient with known cardiac disease [22], but also for those with increased cardiovascular risk due to underlying risk factors. After basic examination and testing (e.g., ECG, treadmill test) the diagnostics should be conducted according to a step-up plan depending on the results (e.g., ultrasound, stress tests, or coronary arteriography). Prior to the trek test, exposure to moderate altitude in a controlled environment (trekking destination with good medical infrastructure or simulated altitude) can be a valuable tool in assessing altitude tolerance with a pre-existing condition. The Borg Scale has recently been reviewed as a valid means to monitor the appropriate level of workload and perceived levels of physical strain for an altitude up to 5000 m [23]. En route self-testing of blood pressure and heart rate (e.g., by smart watch) is essential to avoid an excess in physical workload, especially during the first days at high altitude.

Data show that trekking is possible for most people with pre-existing health conditions and that there is no need to advise them not to go as a matter of principle.

Limitations: Since the study was a field project, there are some limitations. First of all, a recall bias cannot be excluded for sure. Imprecise wording of some questions and optional free-text entries made data evaluation difficult. A certain selection bias is possible since not all trekkers that were contacted en route took part, and maybe health-conscious persons were more interested in the study than others. More limitations to be discussed are language barrier, transcription errors of handwritten answers in the questionnaires, incomplete given data, or self-overestimation of participants, e.g., when asked for weekly hours of physical activity per week. However, since these limitations are typical for any field study in the mountains, the results should be comparable to other studies.

## 5. Conclusions

Every person trekking in the Himalayas should consult a physician prior to departure, ideally assessed by a travel medicine specialist with training and personal experience in high-altitude medicine. The topic “acute mountain sickness” warrants special attention. Travellers with heart disease or with a pronounced cardiovascular risk profile should be presented to an internal medicine professional. Travel plans must be adjusted individually, especially concerning adequate time for acclimatisation. With these and other precautions, trekking at high altitudes is generally possible even with significant pre-existing conditions and can lead to invaluable personal experiences.

## Figures and Tables

**Figure 1 ijerph-19-16388-f001:**
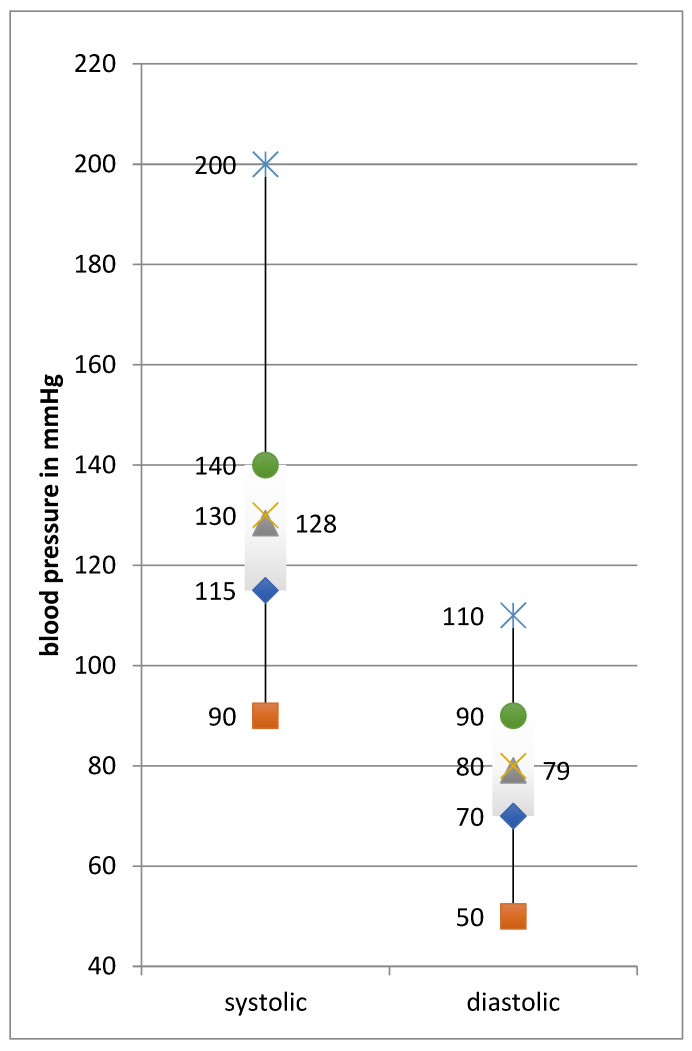
Blood pressure of the collective.

**Figure 2 ijerph-19-16388-f002:**
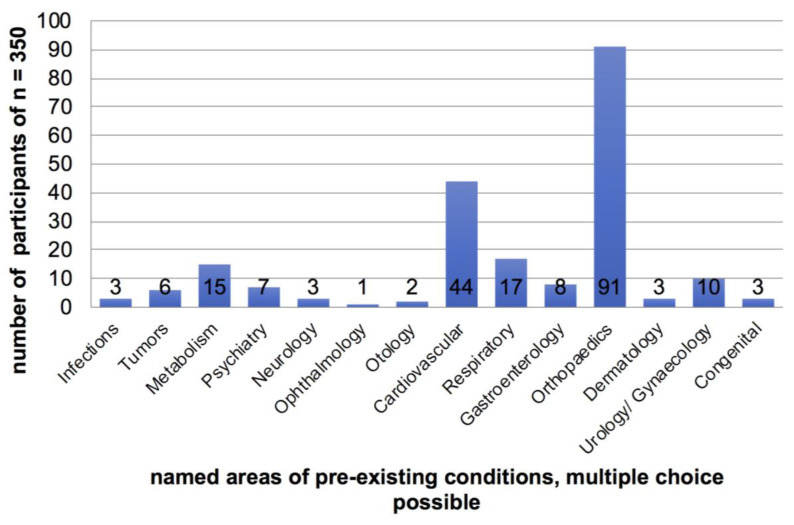
Pre-existing conditions of the collective (*n* = 350).

**Figure 3 ijerph-19-16388-f003:**
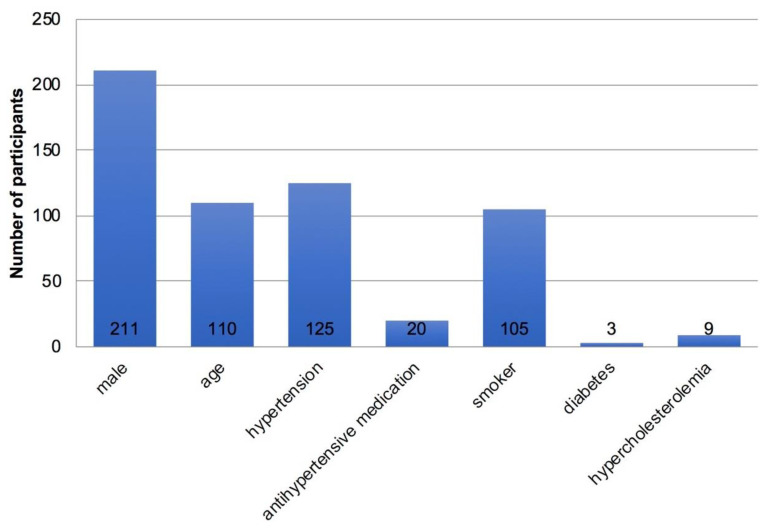
Occurrence of major cardiovascular risk factors (*n* = 350).

**Figure 4 ijerph-19-16388-f004:**
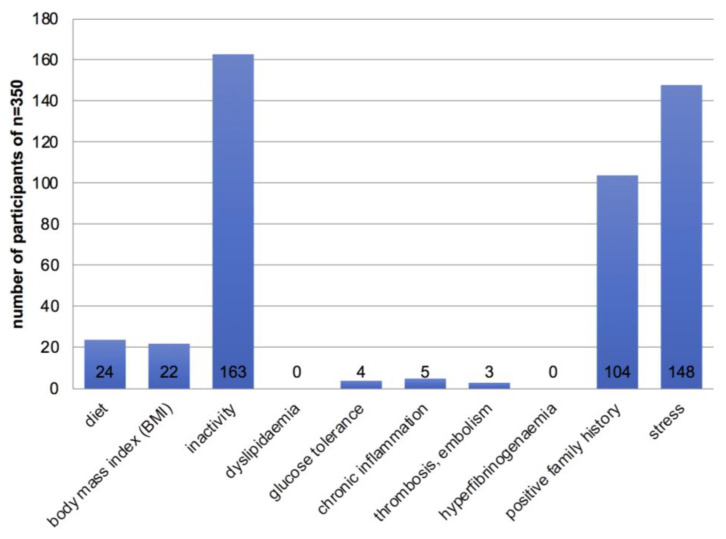
Occurrence of minor cardiovascular risk factors (*n* = 350).

**Table 1 ijerph-19-16388-t001:** Cardiovascular 10-year risk and heart age.

Averaged Parameter	BMI Formula	Cholesterol Formula	Total	Mean Age of the Collective for Comparison
**10-year risk** (**women**)	4.1%	4.6%	4.4%	
**10-year risk** (**men**)	9.0%	7.8%	8.4%	
**10-year risk** (**both**)	7.4%	6.4%	6.9%	
**Heart age** (**women**)	45.0 years	50.1 years	47.6 years	42.7 years
**Heart age** (**men**)	47.4 years	47.7 years	47.6 years	42.2 years
**Heart age** (**both**)	46.6 years	48.7 years	47.7 years	42.7 years
**Number of calculations** (***n***)	274	32		

## Data Availability

Data are archived by the first author and the senior author.

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
