# Peer review of "Cardiovascular Risk Profiles and Pre-Existing Health Conditions of Trekkers in the Solu-Khumbu Region, Nepal"

_ijerph, 2022, doi:10.3390/ijerph192416388_

Round 1
Reviewer 1 Report
Dear Authors
Thank you for the opportunity to review your manuscript with the topic: “Cardiovascular risk profiles and pre-existing health conditions of trekkers in the Solu-Khumbu region, Nepal”
This is a highly topical subject of great relevance that fits in the scope of the International Journal of Environmental Research and Public Health, especially in the planed special edition of IJERPH.
Nevertheless, I have some points that should be addressed bevor a publication in IJERPH can be considered.
My biggest problem is the confusing information about the incidence of high altitude sickness (see below). It gives me the impression that a problem is being exaggerated here. It is even implied that almost 15% of trekkers suffer from the life-threatening forms of HACE and HAPE (L131-133). However, it is not specified which (therapeutic) consequences were derived from this.
The basically good manuscript does not need this dramatization. The focus of this paper according to the title and introduction is not on AMS but on cardiovascular risk. It is possible that this confusion is due to the use of two different questionnaires, with only the LLS stated as such and the other is not described further (L77) Since the focus of this study is not the different quantification possibilities of AMS, I recommend using the usual LLS. In doing so, I consider it legitimate and even useful to transfer the obtained answers to the LLS2018.
The introduction is well written. It explains the background of the study to the reader and leads well to the question of the study. The large number of trekkers interviewed is also a strength of this study.
Major points:
Material and Methods:
Even if the investigations are described basically here, this description is too imprecise to evaluate the results in detail or to repeat this study. It is not stated which pulse oximeter was used or how the blood pressure was determined. Did you measure body weight (what kind of balance) or did you only ask for it, the same with body size? There is no information on how cholesterol values were determined and why this was done in less than 10% of all subjects. Calculation of heart risk and heart age should be done in the same way for the whole group. This means you should use BMI for all participants. If differences occur using cholesterol in the subgroup this can be discussed.
The questionnaire used should be presented. I am wondering how the information’s of Fig 3 were obtained especially “stress”. The language of the questionnaire is not given. Was it clearly understandable for e.g. French people (largest part of the trekkers interviewed).
Please describe how “healthy lifestyle” was defined and what kind of questionnaire you used. Are the results from this questionnaire (Line 79) transferable to the study population (average age 42.7 years) because Lit No 9 deals with young adults?
I see a fundamental problem in recruitment of the subjects for the study, because persons with pre-existing conditions or current illnesses presumably have a higher motivation to participate in a medical examination and this may result in a systematic selection bias with increased incidences of illnesses in the selected study population. How many people were asked to participate in the study and how many of them agreed to participate. This should be discussed in a previously missing section "Limitations of the study".
Results:
I have difficulties with your results concerning AMS. According to the LLS AMS is defined as a score equal or higher of 3 in the presence of headache. A large number of your subjects (25%) suffer from manifest AMS (L125) but only 18% (L120) have headache. 61% of the study population (in the abstract 72%, L30) have one potential symptom of AMS (L124), which symptoms? If they do not reach a score ≥ 3 they do not have AMS. Even if it can be formally correct to speak of symptoms of AMS, e.g. mild headache has no relevance and no therapeutic consequence – and mild headache is no AMS. It is confusing to me that with regard to the actual LLS2018 only 20 persons (5.7%) have AMS (L130) but 19 trekker show signs of HACE and 30 trekker risk factors of HAPE. This seems to me as if dramatized here because apparently no consequences were drawn, which would have been necessary in the presence of these life-threatening types of high altitude illness. At least nothing corresponding is indicated.
Tab 1 shows a total of 306 numbers of calculations with regard to 350 trekkers. It should be explained why 46 are missing. As in less than 10% of the study population cholesterol was measured a BMI calculation of all participants should be presented. In addition the cholesterol calculation can be presented and discussed as additional information.
Orthopedic conditions are the majority of pre-existing condition (Fig 1). In line 165 it is described that pre-existing conditions have worsened during the trekking. Does this include orthopedic conditions, too? As the focus of the manuscript are cardiovascular risks this has to be specified.
It is not describe how blood pressure was measured. I would recommend not to use decimal digits as i.e. riva-rocci method of measurement is not accurate enough.
Fig. 3 How was “stress” quantified
Discussion
The given information of line 207-209 is very important and is almost lost here.
Formally, this result of the questionnaire should be presented in the “Result” section. From this, the recommendations given that profound medical and travel medical advice should be consulted before a trekking stay are derived. This should be discussed in more detail and in a more striking way, since considerable health risks are involved.
L 212 “Signs of AMS” are no AMS. Therefore, the figure of 25% is too dramatic and misleads the reader, who is not familiar with high altitude medicine, to draw the wrong conclusions.
Also, the three references (12, 13, 14) given seem to be all from the same study. It would be desirable to have information on the incidence of cardiovascular events or altitude sickness during trekking in the Everst region.
A section “Limitations of the study” is missing.
Conclusion
I completely agree with this conclusions.
Author Response
see attached file, please!

Reviewer 2 Report
Dear Editor of the International Journal of Environmental Research and Public Health
Thank you for referring the article entitle "Cardiovascular risk profiles and pre-existing health conditions of trekkers in the Solu-Khumbu region, Nepal" to me. My comments can be seen below.
- The introduction section is very short and incoherent and does not provide enough information about the importance of the study. For example, the first and second sentences are not in the right sequence, and what is the cardiovascular risk in similar samples? What does the study literature say? I think the introduction part needs a serious revision.
- The methods section also does not provide detailed information for replication of the study by other researchers. What is the study design (quantitative or qualitative)? Number of subjects? Initial and final sample size? What is the sampling method? Consecutive sampling? What are the entry and exit criteria? Where did they answer the questions? Who collected the data? There are many standard indicators to evaluate cardiovascular health. What is the reason for using a non-standard self-report questionnaire? Because many of the general population greatly underestimate cardiovascular risk or its risk factors. How were anthropometric (e.g. BMI) measurements done? What are the details of standard measuring instruments?
I think the authors should address all these issues using separate headings.
- The discussion section can be strengthened. The conclusion is somewhat scattered and needs to be reported more coherently and based on the main findings. Pointing out the strengths and limitations of the study can also provide useful information for future studies.
Round 2
Reviewer 2 Report
Dear editor,
I am grateful to the authors for addressing my comments. Although I believe the methods section could be improved, the manuscript is acceptable in its current format.
Best